# Real-Time Decoding of an Integrate and Fire Encoder

Shreya Saxena and Munther Dahleh

Department of Electrical Engineering and Computer Sciences
Massachusetts Institute of Technology
Cambridge, MA 02139
{ssaxena,dahleh}@mit.edu

## Abstract

Neuronal encoding models range from the detailed biophysically-based Hodgkin
Huxley model, to the statistical linear time invariant model specifying firing rates
in terms of the extrinsic signal. Decoding the former becomes intractable, while
the latter does not adequately capture the nonlinearities present in the neuronal
encoding system. For use in practical applications, we wish to record the output
of neurons, namely spikes, and decode this signal fast in order to act on this signal,
for example to drive a prosthetic device. Here, we introduce a causal, real-time
decoder of the biophysically-based Integrate and Fire encoding neuron model. We
show that the upper bound of the real-time reconstruction error decreases polyno-
mially in time, and that the $\mathcal{L}_2$ norm of the error is bounded by a constant that
depends on the density of the spikes, as well as the bandwidth and the decay of
the input signal. We numerically validate the effect of these parameters on the
reconstruction error.

## 1 Introduction

One of the most detailed and widely accepted models of the neuron is the Hodgkin Huxley (HH)
model [1]. It is a complex nonlinear model comprising of four differential equations governing
the membrane potential dynamics as well as the dynamics of the sodium, potassium and calcium
currents found in a neuron. We assume in the practical setting that we are recording multiple neurons
using an extracellular electrode, and thus that the observable postprocessed outputs of each neuron
are the time points at which the membrane voltage crosses a threshold, also known as spikes. Even
with complete knowledge of the HH model parameters, it is intractable to decode the extrinsic
signal applied to the neuron given only the spike times. Model reduction techniques are accurate in
certain regimes [2]; theoretical studies have also guaranteed an input-output equivalence between a
multiplicative or additive extrinsic signal applied to the HH model, and the same signal applied to
an Integrate and Fire (IAF) neuron model with variable thresholds [3].

Specifically, take the example of a decoder in a brain machine interface (BMI) device, where the
decoded signal drives a prosthetic limb in order to produce movement. Given the complications
involved in decoding an extrinsic signal using a realistic neuron model, current practices include
decoding using a Kalman filter, which assumes a linear time invariant (LTI) encoding with the ex-
trinsic signal as an input and the firing rate of the neuron as the output [4–6]. Although extremely
tractable for decoding, this approach ignores the nonlinear processing of the extrinsic current by
the neuron. Moreover, assuming firing rates as the output of the neuron averages out the data and
incurs inherent delays in the decoding process. Decoding of spike trains has also been performed
using stochastic jump models such as point process models [7, 8], and we are currently exploring
relationships between these and our work.

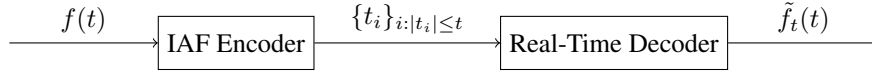

Figure 1: IAF Encoder and a Real-Time Decoder.

We consider a biophysically inspired IAF neuron model with variable thresholds as the encoding model. It has been shown that, given the parameters of the model and given the spikes for all time, a bandlimited signal driving the IAF model can be perfectly reconstructed if the spikes are 'dense enough' [9–11]. This is a Nyquist-type reconstruction formula. However, for this theory to be applicable to a real-time setting, as in the case of BMI, we need a causal real-time decoder that estimates the signal at every time $t$, and an estimate of the time taken for the convergence of the reconstructed signal to the real signal. There have also been some approaches for causal reconstruction of a signal encoded by an IAF encoder, such as in [12]. However, these do not show the convergence of the estimate to the real signal with the advent of time.

In this paper, we introduce a causal real-time decoder (Figure 1) that, given the parameters of the IAF encoding process, provides an estimate of the signal at every time, without the need to wait for a minimum amount of time to start decoding. We show that, under certain conditions on the input signal, the upper bound of the error between the estimated signal and the input signal decreases polynomially in time, leading to perfect reconstruction as $t \rightarrow \infty$, or a bounded error if a finite number of iterations are used. The bounded input bounded output (BIBO) stability of a decoder is extremely important to analyze for the application of a BMI. Here, we show that the $\mathcal{L}_2$ norm of the error is bounded, with an upper bound that depends on the bandwidth of the signal, the density of the spikes, and the decay of the input signal.

We numerically show the utility of the theory developed here. We first provide example reconstructions using the real-time decoder and compare our results with reconstructions obtained using existing methods. We then show the dependence of the decoding error on the properties of the input signal.

The theory and algorithm presented in this paper can be applied to any system that uses an IAF encoding device, for example in pluviometry. We introduce some preliminary definitions in Section 2, and then present our theoretical results in Section 3. We use a model IAF system to numerically simulate the output of an IAF encoder and provide causal real-time reconstruction in Section 4, and end with conclusions in Section 5.

## 2  Preliminaries

We first define the subsets of the $\mathcal{L}_2$ space that we consider. $\mathcal{L}_2^\Omega$ and $\mathcal{L}_{2,\beta}^\Omega$ are defined as the following.

$$\mathcal{L}_2^\Omega = \left\{ f \in \mathcal{L}_2 \mid \hat{f}(\omega) = 0 \ \forall \omega \notin [-\Omega, \Omega] \right\} \tag{1}$$

$$\mathcal{L}_{2,\beta}^\Omega = \left\{ fg_\beta \in \mathcal{L}_2 \mid \hat{f}(\omega) = 0 \ \forall \omega \notin [-\Omega, \Omega] \right\} \tag{2}$$

, where $g_\beta(t) = (1+|t|)^\beta$ and $\hat{f}(\omega) = (\mathcal{F}f)(\omega)$ is the Fourier transform of $f$. We will only consider signals in $\mathcal{L}_{2,\beta}^\Omega$ for $\beta \geq 0$.

Next, we define $\operatorname{sinc}_\Omega(t)$ and $\mathbb{1}_{[a,b]}(t)$, both of which will play an integral part in the reconstruction of signals.

$$\operatorname{sinc}_\Omega(t) = \begin{cases} \frac{\sin(\Omega t)}{\Omega t} & t \neq 0 \\ 1 & t = 0 \end{cases} \tag{3}$$

$$\mathbb{1}_{[a,b]}(t) = \begin{cases} 1 & t \in [a, b] \\ 0 & otherwise \end{cases} \tag{4}$$

Finally, we define the encoding system based on an IAF neuron model; we term this the IAF Encoder. We consider that this model has variable thresholds in its most general form, which may be useful if

it is the result of a model reduction technique such as in [3], or in approaches where $\int_{t_i}^{t_{i+1}} f(\tau)d\tau$ can be calculated through other means, such as in [9]. A typical IAF Encoder is defined in the following way: given the thresholds $\{q_i\}$ where $q_i > 0$ $\forall i$, the spikes $\{t_i\}$ are such that

$$\int_{t_i}^{t_{i+1}} f(\tau)d\tau = \pm q_i \tag{5}$$

This signifies that the encoder outputs a spike at time $t_{i+1}$ every time the integral $\int_{t_i}^{t} f(\tau)d\tau$ reaches the threshold $q_i$ or $-q_i$. We assume that the decoder has knowledge of the value of the integral as well as the time at which the integral was reached. For a physical representation with neurons whose dynamics can faithfully be modeled using IAF neurons, we can imagine two neurons with the same input $f$; one neuron spikes when the positive threshold is reached while the other spikes when the negative threshold is reached. The decoder views the activity of both of these neurons and, with knowledge of the corresponding thresholds, decodes the signal accordingly. We can also take the approach of limiting ourselves to positive $f(t)$. In order to remain general in the following treatment, we assume that we have knowledge of $\left\{ \int_{t_i}^{t_{i+1}} f(\tau)d\tau \right\}$, as well as the corresponding spike times $\{t_i\}$.

## 3 Theoretical Results

The following is a theorem introduced in [11], which was also applied to IAF Encoders in [10,13,14]. We will later use the operators and concepts introduced in this theorem.

**Theorem 1.** *Perfect Reconstruction: Given a sampling set $\{t_i\}_{i \in \mathcal{Z}}$ and the corresponding samples $\int_{t_i}^{t_{i+1}} f(\tau)d\tau$, we can perfectly reconstruct $f \in \mathcal{L}_2^{\Omega}$ if $\sup_{i \in \mathcal{Z}}(t_{i+1} - t_i) = \delta$ for some $\delta < \frac{\pi}{\Omega}$. Moreover, $f$ can be reconstructed iteratively in the following way, such that*

$$\|f - f^k\|_2 \leq \left( \frac{\delta \Omega}{\pi} \right)^{k+1} \|f\|_2 \tag{6}$$

*, and $\lim_{k \to \infty} f^k = f$ in $\mathcal{L}_2$.*

$$f^0 = \mathcal{A}f \tag{7}$$
$$f^1 = (I - \mathcal{A})f^0 + \mathcal{A}f = (I - \mathcal{A})\mathcal{A}f + \mathcal{A}f \tag{8}$$
$$f^k = (I - \mathcal{A})f^{k-1} + \mathcal{A}f = \sum_{n=0}^{k}(I - \mathcal{A})^n \mathcal{A}f \tag{9}$$

*, where the operator $\mathcal{A}f$ is defined as the following.*

$$\mathcal{A}f = \sum_{i=1}^{\infty} \int_{t_i}^{t_{i+1}} f(\tau)d\tau \, \mathrm{sinc}_{\Omega}(t - s_i) \tag{10}$$

*and $s_i = \frac{t_i + t_{i+1}}{2}$, the midpoint of each pair of spikes.*

*Proof.* Provided in [11]. $\square$

The above theorem requires an infinite number of spikes in order to start decoding. However, we would like a real-time decoder that outputs the 'best guess' at every time $t$ in order for us to act on the estimate of the signal. In this paper, we introduce one such decoder; we first provide a high-level description of the real-time decoder, then a recursive algorithm to apply in the practical case, and finally we will provide error bounds for its performance.

**Real-Time Decoder**
At every time $t$, the decoder outputs an estimate of the input signal $\tilde{f}_t(t)$, where $\tilde{f}_t(t)$ is an estimate of the signal calculated using all the spikes from time $0$ to $t$. Since there is no new information between spikes, this is essentially the same as calculating an estimate after every spike $t_i$, $\tilde{f}_{t_i}(t)$, and using this estimate till the next spike, i.e. for time $t \in [t_i, t_{i+1}]$ (see Figure 2).

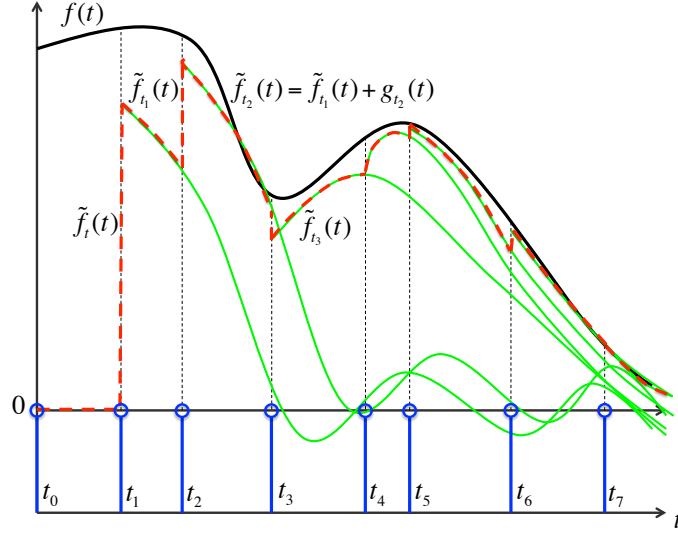

Figure 2: A visualization of the decoding process. The original signal $f(t)$ is shown in black and the spikes $\{t_i\}$ are shown in blue. As each spike $t_i$ arrives, a new estimate $\tilde{f}_{t_i}(t)$ of the signal is formed (shown in green), which is modified after the next spike $t_{i+1}$ by the innovation function $g_{t_{i+1}}$. The output of the decoder $\tilde{f}_t(t) = \sum_{i \in \mathcal{Z}} \tilde{f}_{t_i}(t) \mathbb{1}_{[t_i, t_{i+1})}(t)$ is shown in red.

We will show that we can calculate the estimate after every spike $\tilde{f}_{t_{i+1}}$ as the sum of the previous estimate $\tilde{f}_{t_i}$ and an innovation $g_{t_{i+1}}$. This procedure is captured in the algorithm given in Equations 11 and 12.

**Recursive Algorithm**

$$\tilde{f}_{t_{i+1}}^0 = \tilde{f}_{t_i}^0 + g_{t_{i+1}}^0 \tag{11}$$

$$\tilde{f}_{t_{i+1}}^k = \tilde{f}_{t_i}^k + g_{t_{i+1}}^k = \tilde{f}_{t_i}^k + \left( g_{t_{i+1}}^{k-1} + g_{t_{i+1}}^0 - \mathcal{A}_{t_{i+1}} g_{t_{i+1}}^{k-1} \right) \tag{12}$$

Here, $\tilde{f}_{t_0}^0 = 0$, and $g_{t_{i+1}}^0(t) = \left( \int_{t_i}^{t_{i+1}} f(\tau) d\tau \right) \mathrm{sinc}(t - s_i)$. We denote $\tilde{f}_{t_i}(t) = \lim_{k \to \infty} \tilde{f}_{t_i}^k(t)$ and $g_{t_{i+1}}(t) = \lim_{k \to \infty} g_{t_{i+1}}^k(t)$. We define the operator $\mathcal{A}_T f$ used in Equation 12 as the following.

$$\mathcal{A}_T f = \sum_{i: |t_i| \leq T} \int_{t_i}^{t_{i+1}} f(\tau) d\tau \, \mathrm{sinc}_\Omega(t - s_i) \tag{13}$$

The output of our causal real-time decoder can also be written as $\tilde{f}_t(t) = \sum_{i \in \mathcal{Z}} \tilde{f}_{t_i}(t) \mathbb{1}_{[t_i, t_{i+1})}(t)$. In the case of a decoder that uses a finite number of iterations $K$ at every step, i.e. calculates $\tilde{f}_{t_i}^K$ after every spike $t_i$, the decoded signal is $\tilde{f}_t^K(t) = \sum_{i \in \mathcal{Z}} \tilde{f}_{t_i}^K(t) \mathbb{1}_{[t_i, t_{i+1})}(t)$. $\{\tilde{f}_{t_i}^k\}_k$ are stored after every spike $t_i$, and thus do not need to be recomputed at the arrival of the next spike. Thus, when a new spike arrives at $t_{i+1}$, each $\tilde{f}_{t_i}^k$ can be modified by adding the innovation functions $g_{t_{i+1}}^k$.

Next, we show an upper bound on the error incurred by the decoder.

**Theorem 2.** *Real-time reconstruction: Given a signal $f \in \mathcal{L}_{2,\beta}^\Omega$ passed through an IAF encoder with known thresholds, and given that the spikes satisfy a certain minimum density $\sup_{i \in \mathbb{Z}}(t_{i+1} - t_i) = \delta$ for some $\delta < \frac{\Omega}{\pi}$, we can construct a causal real-time decoder that reconstructs a function $\tilde{f}_t(t)$ using the recursive algorithm in Equations 11 and 12, s.t.*

$$|f(t) - \tilde{f}_t(t)| \leq \frac{c}{1 - \frac{\delta\Omega}{\pi}} \|f\|_{2,\beta} (1 + t)^{-\beta} \tag{14}$$

, where $c$ depends only on $\delta$, $\Omega$ and $\beta$.

*Moreover, if we use a finite number of iterations $K$ at every step, we obtain the following error.*

$$|f(t) - \tilde{f}_t^K(t)| \leq c \frac{1 - \left(\frac{\delta\Omega}{\pi}\right)^{K+1}}{1 - \frac{\delta\Omega}{\pi}} \|f\|_{2,\beta}(1+t)^{-\beta} + \left(\frac{\delta\Omega}{\pi}\right)^{K+1} \frac{1 + \frac{\delta\Omega}{\pi}}{1 - \frac{\delta\Omega}{\pi}} \|f\|_2 \qquad (15)$$

*Proof.* Provided in the Appendix. $\qquad\qquad\square$

Theorem 2 is the main result of this paper. It shows that the upper bound of the real-time reconstruction error using the decoding algorithm in Equations 11 and 12, decreases polynomially as a function of time. This implies that the approximation $\tilde{f}_t(t)$ becomes more and more accurate with the passage of time, and moreover, we can calculate the exact amount of time we would need to record to have a given level of accuracy. Given a maximum allowed error $\epsilon$, these bounds can provide a combination $(t, K)$ that will ensure $|f(t) - \tilde{f}_t^K(t)| \leq \epsilon$ if $f \in \mathcal{L}_{2,\beta}^\Omega$, and if the density constraint is met.

We can further show that the $\mathcal{L}_2$ norm of the reconstruction remains bounded with a bounded input (BIBO stability), by bounding the $\mathcal{L}_2$ norm of the error between the original signal and the reconstruction.

**Corollary 1.** *Bounded $\mathcal{L}_2$ norm: The causal decoder provided in Theorem 2, with the same assumptions and in the case of $K \to \infty$, constructs a signal $\tilde{f}_t(t)$ s.t. the $\mathcal{L}_2$ norm of the error $\|f - \tilde{f}_t\|_2 = \sqrt{\int_0^\infty |f(t) - \tilde{f}_t(t)|^2 dt}$ is bounded: $\|f - \tilde{f}_t\|_2 \leq \frac{c/\sqrt{2\beta - 1}}{1 - \frac{\delta\Omega}{\pi}} \|f\|_{2,\beta}$ where $c$ is the same constant as in Theorem 2.*

*Proof.*

$$\sqrt{\int_0^\infty |f(t) - \tilde{f}_t(t)|^2 dt} \quad \leq \quad \sqrt{\int_0^\infty \left(\frac{c}{1 - \frac{\delta\Omega}{\pi}}\right)^2 \|f\|_{2,\beta}^2 (1+t)^{-2\beta} dt} = \frac{c/\sqrt{2\beta - 1}}{1 - \frac{\delta\Omega}{\pi}} \|f\|_{2,\beta} \quad (16)$$

Here, the first inequality is due to Theorem 2, and all the constants are as defined in the same. $\qquad\square$

***Remark 1***: This result also implies that we have a decay in the root-mean-square (RMS) error, i.e. $\sqrt{\frac{1}{T} \int_0^T |f(t) - \tilde{f}_t(t)|^2 dt} \xrightarrow{T \to \infty} 0$. For the case of a finite number of iterations $K < \infty$, the RMS error converges to a non-zero constant $\left(\frac{\delta\Omega}{\pi}\right)^{K+1} \frac{1 + \frac{\delta\Omega}{\pi}}{1 - \frac{\delta\Omega}{\pi}} \|f\|_2$.

***Remark 2***: The methods used in Corollary 1 also provide a bound on the error in the weighted $\mathcal{L}_2$ norm, i.e. $\|f - \tilde{f}\|_{2,\beta} \leq \frac{c/\sqrt{\beta - 1}}{1 - \frac{\delta\Omega}{\pi}} \|f\|_{2,\beta}$ for $\beta \geq 2$, which may be a more intuitive form to use for a subsequent stability analysis.

## 4  Numerical Simulations

We simulated signals $f(t)$ of the following form, for $t \in [0, 100]$, using a stepsize of $10^{-2}$.

$$f(t) = \frac{\sum_{i=1}^{50} w_k \left(\text{sinc}_\Omega (t - d_k)\right)^\beta}{\sum_{i=1}^{50} w_k} \qquad (17)$$

Here, the $w_k$'s and $d_k$'s were picked uniformly at random from the interval $[0, 1]$ and $[0, 100]$ respectively. Note that $f \in \mathcal{L}_{2,\beta}^{\beta\Omega}$. All simulations were performed using MATLAB R2014a. For each simulation experiment, at every time $t$ we decoded using only the spikes before time $t$.

We first provide example reconstructions using the Real-Time Decoder for four signals in Figure 3, using constant thresholds, i.e. $q_i = q \; \forall i$. We compare our results to those obtained using a Linear Firing Rate (FR) Decoder, i.e. we let the reconstructed signal be a linear function of the number of spikes in the past $\Delta$ seconds, $\Delta$ being the window size. We can see that there is a delay in the reconstruction with this decoding approach. Moreover, the reconstruction is not as accurate as that using the Real-Time Decoder.

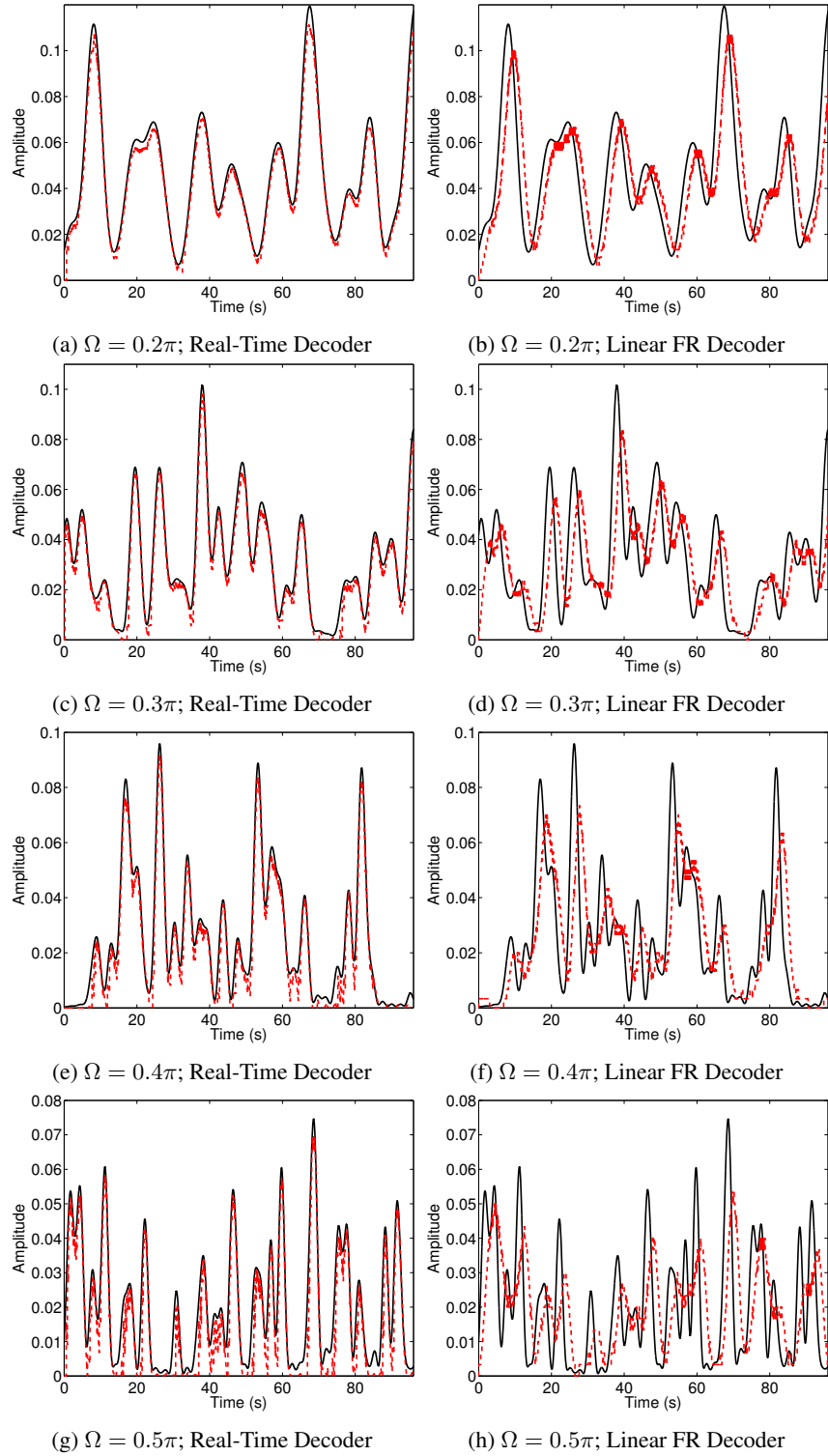

(a) $\Omega = 0.2\pi$; Real-Time Decoder

(b) $\Omega = 0.2\pi$; Linear FR Decoder

(c) $\Omega = 0.3\pi$; Real-Time Decoder

(d) $\Omega = 0.3\pi$; Linear FR Decoder

(e) $\Omega = 0.4\pi$; Real-Time Decoder

(f) $\Omega = 0.4\pi$; Linear FR Decoder

(g) $\Omega = 0.5\pi$; Real-Time Decoder

(h) $\Omega = 0.5\pi$; Linear FR Decoder

Figure 3: (a,c,e,g) Four example reconstructions using the Real-Time Decoder, with the original signal $f(t)$ in black solid and the reconstructed signal $\tilde{f}_t(t)$ in red dashed lines. Here, $[\beta, K] = [2, 500]$, and $q_i = 0.01\ \forall i$. (b,d,f,h) The same signal was decoded using a Linear Firing Rate (FR) Decoder. A window size of $\Delta = 3$s was used.

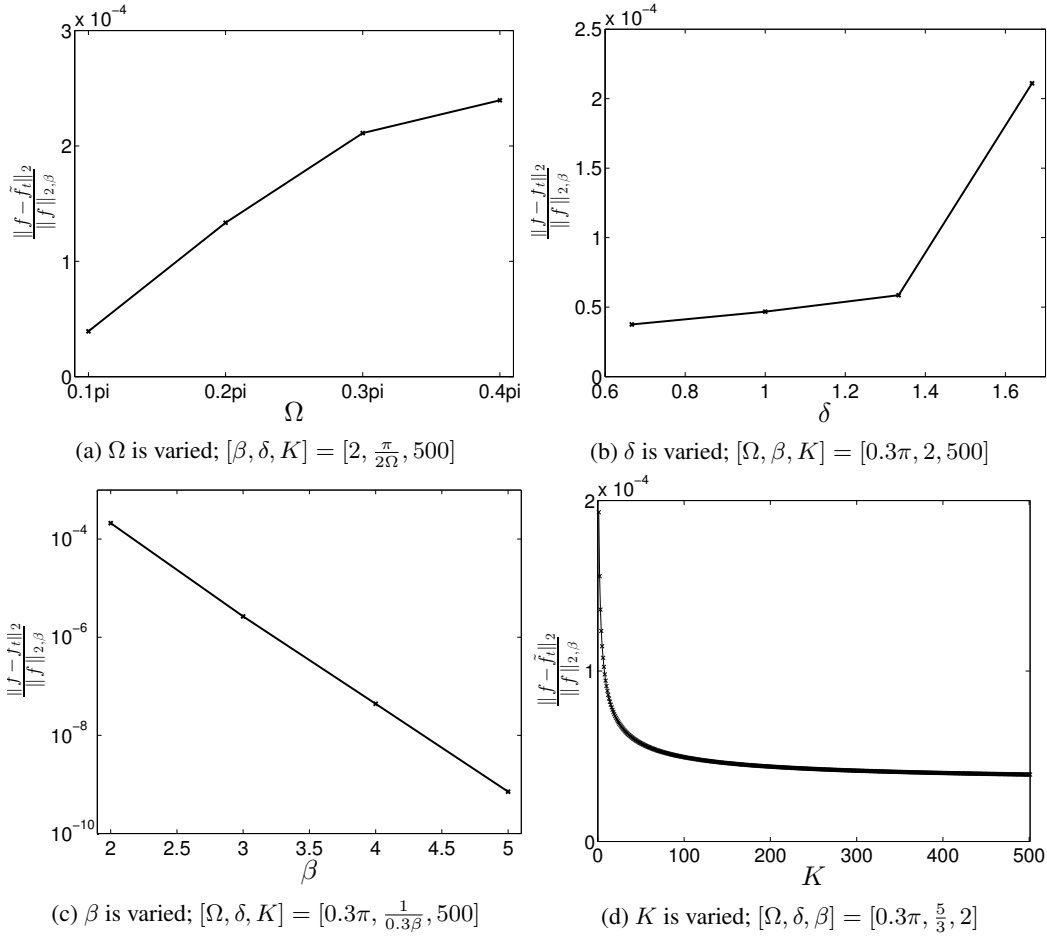

(a) $\Omega$ is varied; $[\beta, \delta, K] = [2, \frac{\pi}{2\Omega}, 500]$

(b) $\delta$ is varied; $[\Omega, \beta, K] = [0.3\pi, 2, 500]$

(c) $\beta$ is varied; $[\Omega, \delta, K] = [0.3\pi, \frac{1}{0.3\beta}, 500]$

(d) $K$ is varied; $[\Omega, \delta, \beta] = [0.3\pi, \frac{5}{3}, 2]$

Figure 4: Average error for 20 different signals while varying different parameters.

Next, we show the decay of the real-time error by averaging out the error for 20 different input signals, while varying certain parameters, namely $\Omega$, $\beta$, $\delta$ and $K$ (Figure 4). The thresholds $q_i$ were chosen to be constant a priori, but were reduced to satisfy the density constraint wherever necessary. According to Equation 14 (including the effect of the constant $c$), the error should decrease as $\Omega$ is decreased. We see this effect in the simulation study in Figure 4a. For these simulations, we chose $\delta$ such that $\frac{\delta\Omega}{\pi} < 1$, thus $\delta$ was decreasing as $\Omega$ increased; however, the effect of the increasing $\Omega$ dominated in this case.

In Figure 4b we see that increasing $\delta$ while keeping the bandwidth constant does indeed increase the error, thus the algorithm is sensitive to the density of the spikes. In this figure, all the values of $\delta$ satisfy the density constraint, i.e. $\frac{\delta\Omega}{\pi} < 1$.

Increasing $\beta$ is seen to have a large effect, as seen in Figure 4c: the error decreases polynomially in $\beta$ (note the log scale on the y-axis). Although increasing $\beta$ in our simulations also increased the bandwidth of the signal, the faster decay had a larger effect on the error than the change in bandwidth.

In Figure 4d, the effect of increasing $K$ is apparent; however, this error flattens out for large values of $K$, showing convergence of the algorithm.

# 5  Conclusions

We provide a real-time decoder to reconstruct a signal $f \in \mathcal{L}_{2,\beta}^{\Omega}$ encoded by an IAF encoder. Under Nyquist-type spike density conditions, we show that the reconstructed signal $\tilde{f}_t(t)$ converges to $f(t)$ polynomially in time, or with a fixed error that depends on the computation power used to reconstruct the function. Moreover, we get a lower error as the spike density increases, i.e. we get better results if we have more spikes. Decreasing the bandwidth or increasing the decay of the signal both lead to a decrease in the error, corroborated by the numerical simulations. This decoder also outperforms the linear decoder that acts on the firing rate of the neuron. However, the main utility of this decoder is that it comes with verifiable bounds on the error of decoding as we record more spikes.

There is a severe need in the BMI community for considering error bounds while decoding signals from the brain. For example, in the case where the reconstructed signal is driving a prosthetic, we are usually placing the decoder and machine in an inherent feedback loop (where the feedback is visual in this case). A stability analysis of this feedback loop includes calculating a bound on the error incurred by the decoding process, which is the first step for the construction of a device that robustly tracks agile maneuvers. In this paper, we provide an upper bound on the error incurred by the real-time decoding process, which can be used along with concepts in robust control theory to provide sufficient conditions on the prosthetic and feedback system in order to ensure stability [15–17].

### Acknowledgments

Research supported by the National Science Foundation's Emerging Frontiers in Research and Innovation Grant (1137237).

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
