[Supplementary Material]

# Real-Time Decoding of an Integrate and Fire Encoder

Shreya Saxena and Munther Dahleh

Department of Electrical Engineering and Computer Sciences
Massachusetts Institute of Technology
Cambridge, MA 02139
{ssaxena,dahleh}@mit.edu

## Appendix

We present Theorem 2 in the main text again, and provide the proof here.

**Theorem 2.** *Real-time reconstruction: Given a signal $f \in \mathcal{L}_{2,\beta}^{\Omega}$ passed through an IAF encoder with known thresholds, and given that the spikes satisfy a certain minimum density $\sup_{i \in \mathbb{Z}}(t_{i+1} - t_i) = \delta$ for some $\delta < \frac{\Omega}{\pi}$, we can construct a causal real-time decoder that reconstructs a function $\tilde{f}_t(t)$ using the recursive algorithm in Equations 11 and 12, s.t.*

$$|f(t) - \tilde{f}_t(t)| \leq \frac{c}{1 - \frac{\delta\Omega}{\pi}} \|f\|_{2,\beta}(1 + t)^{-\beta} \tag{1}$$

*, where $c$ depends only on $\delta$, $\Omega$ and $\beta$.*
*Moreover, if we use a finite number of iterations $K$ at every step, we obtain the following error.*

$$|f(t) - \tilde{f}_t^K(t)| \leq c \frac{1 - \left(\frac{\delta\Omega}{\pi}\right)^{K+1}}{1 - \frac{\delta\Omega}{\pi}} \|f\|_{2,\beta}(1 + t)^{-\beta} + \left(\frac{\delta\Omega}{\pi}\right)^{K+1} \frac{1 + \frac{\delta\Omega}{\pi}}{1 - \frac{\delta\Omega}{\pi}} \|f\|_2 \tag{2}$$

*Proof.* We start with a few preliminaries, and the definition of a constant $c$ that will be used later in the proof. These preliminary notions are also provided in [1]. We define a local maximum function on $f$:

$$f^{\#}(t) = \sup_{|u| \leq \delta} |f(t + u)| \tag{3}$$

Note the following two properties of $f^{\#}(t)$ for $f \in \mathcal{L}_{2,\beta}^{\Omega}$, with a function $p_{\alpha}(t)$ such that $\hat{p}_{\alpha}(\omega) = 1$ for $\omega \in [-\Omega, \Omega]$, and $p_{\alpha} \in \mathcal{L}_{1,\alpha}$, for some $\alpha \geq \beta$. Here, $*$ denotes the convolution operator.

$$|\sum_{i=1}^{\infty} f(s_i) \mathbb{1}_{[t_i, t_{i+1}]}(t)| \leq f^{\#}(t) \text{ pointwise} \tag{4}$$

$$f^{\#}(t) = (f * p_{\alpha})^{\#}(t) \leq (|f| * p_{\alpha}^{\#})(t) \tag{5}$$

As a consequence of equation (5), we obtain the following bound on the $\mathcal{L}_{2,\beta}$ norm of the local maximum function (3), using a function $p_{\alpha}(t)$ as described above.

$$\|f^{\#}\|_{2,\beta} \leq \inf_{p_{\alpha}} \|p_{\alpha}^{\#}\|_{1,\alpha} \|f\|_{2,\beta} \tag{6}$$

We denote $\inf_{p_{\alpha}} \|p_{\alpha}^{\#}\|_{1,\alpha}$ for some $\alpha \geq \beta$ by $c$, which depends only on $\delta$, $\Omega$ and $\beta$.

We now bound the $\mathcal{L}_2$ norm of the error incurred using a decoder acting on all the spikes in a finite time period $T$, i.e. $\{t_i\}_{i:|t_i| \leq T}$, and show that this error is decaying in $T$.

We consider that after the first approximation $\mathcal{A}_T f$ with a finite number of spikes, we can construct $\{t_i\}_{i:|t_i|>T}$ such that $\sup_{i:|t_i|>T}(t_{i+1} - t_i)$ is less than the required $\delta$. Thus we can construct an

operator $\mathcal{A}$ as long as it is not acting directly on $f$, where $\mathcal{A}$ is defined as in Theorem 1 in the main text. The adjoint operators of $\mathcal{A}$ and $\mathcal{A}_T$ for $f \in \mathcal{L}_2^\Omega$ are $\mathcal{A}^*$ and $\mathcal{A}_T^*$ respectively.

$$\mathcal{A}^* f = \sum_{i \in \mathbb{Z}} f(s_i)(\text{sinc}_\Omega * \mathbb{1}_{[t_i, t_{i+1}]}) \tag{7}$$

$$\mathcal{A}_T^* f = \sum_{i:|t_i| \le T} f(s_i)(\text{sinc}_\Omega * \mathbb{1}_{[t_i, t_{i+1}]}) \tag{8}$$

Equation 7 is shown in [2], and Equation 8 follows similarly.

We first define $\tilde{f}_T$ as the result of the following iteration, i.e. $\tilde{f}_T = \lim_{k \to \infty} \tilde{f}_T^k$.

$$\tilde{f}_T^0 = \mathcal{A}_T f \tag{9}$$

$$\tilde{f}_T^1 = (I - \mathcal{A})\tilde{f}_T^0 + \mathcal{A}_T f = (I - \mathcal{A})\mathcal{A}_T f + \mathcal{A}_T f \tag{10}$$

$$\tilde{f}_T^k = (I - \mathcal{A})\tilde{f}_T^{k-1} + \mathcal{A}_T f = \sum_{n=0}^{k}(I - \mathcal{A})^n \mathcal{A}_T f \tag{11}$$

To derive $\|f - \tilde{f}_T\|_2$ for $f \in \mathcal{L}_2^\Omega$, we first note that the error incurred using a finite number of spikes is the same as the error in the adjoint space, i.e. $\|f - \tilde{f}_T\|_2 = \|f - \sum_{n=0}^{\infty}(I - \mathcal{A})^n \mathcal{A}_T f\|_2 = \|f - \sum_{n=0}^{\infty}(I - \mathcal{A}^*)^n \mathcal{A}_T^* f\|_2$

We can thus work exclusively with the adjoint operators $\mathcal{A}^* f$ and $\mathcal{A}_T^* f$ in order to derive $\|f - \tilde{f}_T\|_2$ [3].

$$\|f - \tilde{f}_T\|_2 = \left\| \sum_{n=0}^{\infty}(I - \mathcal{A}^*)^n(\mathcal{A}^* f - \mathcal{A}_T^* f) \right\|_2 \tag{12}$$

$$\le \sum_{n=0}^{\infty} \left\| (I - \mathcal{A}^*)^n(\mathcal{A}^* f - \mathcal{A}_T^* f) \right\|_2 \tag{13}$$

$$\le \sum_{n=0}^{\infty} \left(\frac{\delta\Omega}{\pi}\right)^{n+1} \left\| \mathcal{A}^* f - \mathcal{A}_T^* f \right\|_2 \tag{14}$$

$$= \frac{1}{1 - \frac{\delta\Omega}{\pi}} \left\| \sum_{i:|t_i|>T} f(s_i)\mathbb{1}_{[t_i,t_{i+1}]} * \text{sinc}_\Omega \right\|_2 \tag{15}$$

$$\le \frac{1}{1 - \frac{\delta\Omega}{\pi}} \left\| \sum_{i:|t_i|>T} f(s_i)\mathbb{1}_{[t_i,t_{i+1}]} \right\|_2 \tag{16}$$

$$\le \frac{1}{1 - \frac{\delta\Omega}{\pi}} \left\| f^{\#} \mathbb{1}_{\mathbb{R}\setminus[-T,T]} \right\|_2 \quad (using\ (5)) \tag{17}$$

$$= \frac{1}{1 - \frac{\delta\Omega}{\pi}} \left\| f^{\#} \mathbb{1}_{\mathbb{R}\setminus[-T,T]}(1 + |t|)^{\beta}(1 + |t|)^{-\beta} \right\|_2 \tag{18}$$

$$\le \frac{1}{1 - \frac{\delta\Omega}{\pi}} \left\| f^{\#} \right\|_{2,\beta} \sup_{t \in \mathbb{R}\setminus[-T,T]} (1 + |t|)^{-\beta} \tag{19}$$

$$\le \frac{c}{1 - \frac{\delta\Omega}{\pi}} \|f\|_{2,\beta} (1 + T)^{-\beta} \quad (using\ (6)) \tag{20}$$

, where $c$ depends only on $\delta$, $\Omega$ and $\beta$.

Now, only using a finite number of iterations, we have the following error bound.

$$\|f - \tilde{f}_T^K\|_2 = \left\| \sum_{n=0}^{\infty}(I - \mathcal{A}^*)^n \mathcal{A}^* f - \sum_{n=0}^{K}(I - \mathcal{A}^*)^n \mathcal{A}_T^* f \right\|_2 \tag{21}$$

$$\le \left\| \sum_{n=0}^{K}(I - \mathcal{A}^*)^n(\mathcal{A}^* f - \mathcal{A}_T^* f) \right\|_2 + \left\| \sum_{n=K+1}^{\infty}(I - \mathcal{A}^*)^n \mathcal{A}^* f \right\|_2 \tag{22}$$

$$\le c\frac{1 - \left(\frac{\delta\Omega}{\pi}\right)^{K+1}}{1 - \frac{\delta\Omega}{\pi}} \|f\|_{2,\beta}(1 + T)^{-\beta} + \left(\frac{\delta\Omega}{\pi}\right)^{K+1} \frac{1 + \frac{\delta\Omega}{\pi}}{1 - \frac{\delta\Omega}{\pi}} \|f\|_2 \tag{23}$$

We now construct $\tilde{f}_t(t)$ using spikes $\{t_i\}_{i:|t_i|\leq t}$ at every time $t$. Thus, at every time $t$, we have a causal decoder that uses all spikes that have already occurred. We bound the error at every time $t$ as the following.

$$
\begin{align}
|f(t) - \tilde{f}_t(t)| &\leq \sup_{\tau \in \mathbb{R}} |f(\tau) - \tilde{f}_t(\tau)| \tag{24} \\
&\leq \|f - \tilde{f}_t\|_2 \tag{25} \\
&\leq \frac{c}{1 - \frac{\delta\Omega}{\pi}} \|f\|_{2,\beta} \, (1+t)^{-\beta} \tag{26}
\end{align}
$$

Here, we used the fact that $\|x\|_\infty \leq \|x\|_2 \; \forall x \in \mathcal{L}_2$ for the inequality in Equation 25, and Equation 20 for the inequality in Equation 26. The proof for Equation 2 follows similarly from Equation 23. We note that as a new spike $t_{i+1}$ arrives, we can calculate the new estimate as a function of the old estimate due to the following.

$$
\tilde{f}^0_{t_{i+1}} = \mathcal{A}_{t_{i+1}} f = \mathcal{A}_{t_i} f + \left( \int_{t_i}^{t_{i+1}} f(\tau)d\tau \right) \mathrm{sinc}(t - s_i) = \tilde{f}^0_{t_i} + g^0_{t_{i+1}} \tag{27}
$$

We can carry the term $g^0_{t_{i+1}}$ forward, and track its effect on future iterations to calculate $g^k_{t_{i+1}}$ as a function of $g^{k-1}_{t_{i+1}}$, to obtain Equation 12 in the main text. $\qquad\square$