[Reviews · NeurIPS 2014]

Submitted by Assigned_Reviewer_20

The paper introduces a model and algorithms for decoding spikes from an integrate and fire decoder. Theorems that are providing upper bounds on the reconstruction error are given and the parameters and and error of the model are studied.

The difference to prior work by the Lazar group does not become clear from the ms to the reviewer. Potentially the contribution comes from this lab. The simulations and numerical studies are on the weaker side. What I do not like is that the authors build up an expectation that their methods may be used in BMI decoding and then do not show any data.
As a theoretical contribution, I consider it solid, but as said above, the margin to prior work does not become clear.
Summary: A theoretical contribution for decoding spikes from an integrate and fire decoder including an upper bound of the reconstruction error. Margin size to existing work unclear, weaker simulations.

Submitted by Assigned_Reviewer_31

The authors introduced a recursive decoder for Integrate and Fire (IAF) encoders to reconstruct the input signal. They theoretically derived the upper bound of the reconstruction error, and moreover, its rate of convergence to the real input as a function of time. This asymptotic convergent analysis is novel, important result. Overall, the work is an interesting, original application of the irregular-time sampling theory that could potentially benefit the construction of Brain-machine interfaces.

The article is technically strong, but a bit difficult to read. It'll be great if the authors can provide better intuitions behind their mathematical derivations.
Summary: It's not clearly explained why the authors want to construct recursive decoder. Is this inspired by biophysical reality of the neural system? Or is it because of mathematical convenience or ease of implementation? It'll be helpful to carefully explain the rationale.

Submitted by Assigned_Reviewer_37

This manuscript describes a method for an Integrate and Fire decoder that allows for, based on an assumption of the encoding parameters, real-time decoding of the underlying signal.

The manuscript is well written and the derivations appear to be sound and well formulated. The presented idea is intended to be a building block for a reliably converging decoder based on the IAF neuron model. Numerical simulations in the manuscript provide insights in the potential quality of the decoder in realistic scenarios, like BMI.
Summary: The work is building up on a sound theoretical framework and is refining previous work of implementing a reliable decoder based on the model assumptions. Unfortunately, I am not an expert in IAF decoders but it appears to me that the work is incremental.
Author Feedback
Author rebuttal: We thank the reviewers for their comments and suggestions, and address their issues together in the following.

Assigned_Reviewer_20: “The difference to prior work by the Lazar group does not become clear from the ms to the reviewer.”
Assigned_Reviewer_37: “… it appears to me that the work is incremental.”

The manuscript introduces a causal real-time decoder of the spikes emanating from an Integrate and Fire (IAF) encoder. Moreover, it provides clear bounds on the accuracy of the reconstruction as a function of the amount of time recorded, which has not been previously addressed in literature.

Lazar et al. only consider a non-causal decoder in the references provided in the manuscript. These works focus on reconstructing the sampled signal as it arrives in a batch format, and do not provide an updated estimate after every spike as we do in this manuscript. As it currently stands, previously developed theory is not directly applicable to the many systems that require real-time decoding, such as Brain Machine Interfaces (BMIs). Specifically for these systems, we need a causal estimate of the signal as the spikes are recorded to provide a real-time estimate of the signal in order to progressively improve the accuracy of the action being performed, instead of incurring delays by having a batch processing of the signal.

Moreover, in addition to providing an estimate at every time, we provide an idea of how much time it would take in order to have an accurate estimate of the movement. The real-time tracking of the error enables us to have a bound on the open loop error, and if we implement this system in feedback, we can reduce this error to an even greater extent. We also show that we converge to the correct estimate, that is, the upper bound on the error decreases as a function of time.

Although we are building upon excellent work by Lazar et al. and Feichtinger et al., we believe that the results detailed in this manuscript are more relevant to the BMI community. There is a potential to drive further research by neuroscientists towards decoding while taking into account a dynamical model of the encoder. As such, we consider NIPS to be the ideal platform for this endeavor.

Assigned_Reviewer_20:
“The simulations and numerical studies are on the weaker side.”
“… the authors build up an expectation that their methods may be used in BMI decoding and then do not show any data.”

Although we have tried to show the utility and results of the decoding, we accept that the simulations can be improved upon. To this end, we can provide more elaborate simulation results in the final manuscript, specifically improving both Figure 3 and Figure 4. We have also carried out a more detailed simulation which involves learning the IAF from synthesized data first, then comparing the reconstruction between our real-time IAF decoder and a linear filtering based on an encoding model that takes in the movement signal and outputs firing rates, as implemented in [1]. We can show that our IAF decoder outperforms the linear filter. Moreover, by construction, a linear filter using firing rates incurs delays, since the firing rates are calculated over a time window.
The simulations show the advantages of using our decoder in an experimental setting.

Assigned_Reviewer_31:
“It's not clearly explained why the authors want to construct recursive decoder.”
The utility of the recursive decoder is to show the convergence of the error bounds. It is also computationally less expensive to implement a recursive decoder as we do not have to save previously calculated values.

Assigned_Reviewer_31:
“… a bit difficult to read.”
We can simplify the language throughout the manuscript, and highlight the aspects of the theoretical results that experimentalists in the BMI community might appreciate, for example, a tradeoff between accuracy and movement speed.

As a concrete example, we would add the following paragraph between lines 305 and 306, i.e. after Theorem 2, in the manuscript.
“This implies that the approximation \tilde f_t(t) becomes more and more accurate with the passage of time, and moreover, we can calculate the exact amount of time we would need to record to have a given level of accuracy.”

References:
[1] Serruya, M. D., Hatsopoulos, N. G., Paninski, L., Fellows, M. R., and Donoghue, J. P. (2002). Brain-Machine Interface: Instant neural control of a movement signal. Nature, (416):141-142.